# Novel Models of *Streptococcus canis* Colonization and Disease Reveal Modest Contributions of M-Like (SCM) Protein

**DOI:** 10.3390/microorganisms9010183

**Published:** 2021-01-16

**Authors:** Ingrid Cornax, Jacob Zulk, Joshua Olson, Marcus Fulde, Victor Nizet, Kathryn A Patras

**Affiliations:** 1Department of Pediatrics, UC San Diego, La Jolla, CA 92093, USA; icornax@its.jnj.com (I.C.); jaolson@health.ucsd.edu (J.O.); vnizet@health.ucsd.edu (V.N.); 2Department of Molecular Virology and Microbiology, Baylor College of Medicine, 1 Baylor Plaza, Houston, TX 77030, USA; jacob.zulk@bcm.edu; 3Institute of Microbiology and Epizootics, Centre of Infection Medicine, Freie Universität Berlin, 14163 Berlin, Germany; Marcus.Fulde@fu-berlin.de; 4Skaggs School of Pharmacy and Pharmaceutical Sciences, UC San Diego, La Jolla, CA 92093, USA

**Keywords:** *Streptococcus canis*, M protein, virulence factor, innate immunity, vaginal colonization

## Abstract

*Streptococcus canis* is a common colonizing bacterium of the urogenital tract of cats and dogs that can also cause invasive disease in these animal populations and in humans. Although the virulence mechanisms of *S. canis* are not well-characterized, an M-like protein, SCM, has recently identified been as a potential virulence factor. SCM is a surface-associated protein that binds to host plasminogen and IgGs suggesting its possible importance in host-pathogen interactions. In this study, we developed in vitro and ex vivo blood component models and murine models of *S. canis* vaginal colonization, systemic infection, and dermal infection to compare the virulence potential of the zoonotic *S. canis* vaginal isolate G361 and its isogenic SCM-deficient mutant (G361∆*scm*). We found that while *S. canis* establishes vaginal colonization and causes invasive disease in vivo, the contribution of the SCM protein to virulence phenotypes in these models is modest. We conclude that SCM is dispensable for invasive disease in murine models and for resistance to human blood components ex vivo, but may contribute to mucosal persistence, highlighting a potential contribution to the recently appreciated genetic diversity of SCM across strains and hosts.

## 1. Introduction

*Streptococcus canis* is a Gram-positive beta-hemolytic group G *Streptococcus* that colonizes the epithelial, respiratory, gastrointestinal, and urogenital surfaces of cats and dogs [1,2,3]. Officially named a species in 1986, *S. canis* is well-recognized in veterinary medicine for causing a variety of invasive diseases across domestic animal species including sepsis, necrotizing fasciitis, urinary tract infection, ulcerative keratitis, and mastitis [4,5,6,7,8,9]. Similar *S. canis* colonization and disease manifestations have been reported in wild animal populations [10,11,12]. Since its first description as a zoonotic agent in 1996 [13], human cases of *S. canis*-mediated endocarditis, septicemia, cellulitis, and periprosthetic joint infection have been reported [14,15,16,17,18,19,20,21]. A retrospective study identified *S. canis* as the causative agent in ~1% of human streptococcal infections, however, given the reliance of Lancefield classification for group G *Streptococcus* identification without further speciation, coupled with close interactions between humans and companion animals, it is likely that *S. canis* human infections are underestimated [22,23].

The genetic diversity and molecular pathogenesis of *S. canis* is being actively explored. There are currently more than 50 multi-locus sequence types (MLST) and 20 genomes for *S. canis* [23,24]. The host immune response to *S. canis* is not well-described, yet the pyogenic nature of many *S. canis* soft tissue infections suggest neutrophils and macrophages may be involved. To date, knowledge of *S. canis* virulence factors remains limited, and is largely extrapolated from genetic similarities to the widely-studied group A *Streptococcus* (*S. pyogenes*) [12]. Similar to *S. pyogenes*, *S. canis* possesses an arginine deiminase system [25], a streptolysin O orthologue [26], lysogenic bacteriophage [27], and an M-like protein termed SCM (or SPASc) [28], which is currently the best characterized among candidate *S. canis* virulence factors [29].

In *S. pyogenes*, the M protein, which is genetically diverse with more than 200 *emm* types, serves multiple roles in pathogenesis and immune evasion [30]. Likewise, *S. canis* SCM displays genetic heterogeneity. There are currently 15 SCM types divided into group I and II alleles [23]. SCM is a fibrillar surface protein [26] which binds plasminogen [29] and the Fc region of IgGs from multiple species including human, dog, cat, and mouse [31]. We recently demonstrated that SCM self-interactions facilitate bacterial aggregation and that SCM interactions with IgG initiate formation of protein complexes in human plasma [32]. However, the contribution of SCM, and impact of its allelic variability, to *S. canis* colonization and virulence remains undefined.

In this study, we incorporated in vitro and ex vivo human blood component models, together with murine models of *S. canis* vaginal colonization, systemic infection, and dermal infection to broadly characterize the virulence potential of the zoonotic *S. canis* vaginal isolate G361, originally isolated from a 40-year-old female, and its isogenic SCM-deficient mutant (G361∆*scm*).

## 2. Materials and Methods

### 2.1. Bacterial Strains, Growth Conditions, and In Vitro Phenotyping Assays

Bacterial strains used in this study include *Streptococcus canis* human vaginal wildtype (WT) isolate G361 [33] and isogenic *scm*-targeted insertional mutant G361∆*scm* [32], *Streptococcus pyogenes* human invasive isolate 5448 M1 and isogenic Δ*emm1* [34], and *Streptococcus agalactiae* human meningeal isolate COH1 (ATCC BAA-1176). All bacterial strains were grown to stationary phase in Todd-Hewitt broth (THB, Hardy Diagnostics), or THB agar plates, at 37 °C without shaking. Erythromycin (5 µg/mL) was added to G361∆*scm* to retain the plasmid insertion. Cultures were diluted in fresh THB and incubated at 37 °C until mid-logarithmic phase (defined as OD_600_ = 0.4). For growth curves, stationary cultures were diluted to OD_600_ = 0.1 in either fresh THB or RPMI-1640 (Gibco) and incubated at 37 °C for 3 h with optical density measured every 30 min. To assess hemolytic activity, WT G361 and G361∆*scm* were grown overnight and 10 µL was spotted onto blood agar plates (TSA with 5% sheep blood, Thermo Scientific) and incubated for 24 h at 37 °C with 5% CO_2_. For minimum inhibitory concentrations (MIC), mid-log phase *S. canis* was diluted 1:100 in THB and 100 µL was added to 96-well microtiter plates. Hydrogen peroxide or hypochlorite (1.5-fold dilution series, final concentrations tested 0–3.5 mM and 0–0.34 mM respectively) was diluted in THB and 100 µL was added to the bacterial plates (200 µL total). The plates were then incubated at 37 °C for 18 h and OD_600_ was measured to determine MIC values (calculated as 90% reduction in OD_600_ from *S. canis* only controls).

### 2.2. Biofilm Formation

*S. canis* and *S. pyogenes* biofilms were assessed using methods adapted from previous work [35]. Briefly, stationary cultures were diluted in THB or RPMI-1640 to OD_600_ = 0.1, and 200 µL added to tissue culture-treated 96-well plates. Biofilms were allowed to form for 48 h at 37 °C without shaking. After washing 3X with PBS, biofilms were stained with 1μM SYTO 13 nucleic acid stain (Invitrogen) for 30 min in the dark. Biofilms were then washed 3X with PBS and quantified by measuring fluorescence at OD_485_/OD_520_ on an Infinite 200 Pro (Tecan) plate reader. Fluorescent images of biofilms were also collected using an Echo Revolve microscope at 100X magnification.

### 2.3. Mammalian Cell Lines and Growth Conditions

Canine macrophage-like cells (DH82), immortalized human vaginal epithelial cells (VK2/E6E7), and human monocyte cell line (THP-1) were acquired from the American Type Culture Collection (ATCC CRL-10389, ATCC CRL-2616, and ATCC TIB-202 respectively). HEK-Blue IL-1β cells (Cat# hkb-il1b) were purchased from InvivoGen. DH82 cells were cultured in Eagle’s Minimum Essential Medium (EMEM) (Gibco) + 15% FBS (heat inactivated). VK2 cells were cultured in keratinocyte serum-free medium (KSFM) (Gibco) with 0.5 ng/mL human recombinant epidermal growth factor and 0.05 mg/mL bovine pituitary extract. THP-1 cells were grown in suspension in the following media: RPMI-1640 (Gibco) + 10% FBS (heat inactivated) + 10 mM HEPES + 1 mM sodium pyruvate + 4500 mg/L glucose + 1500 μg/mL sodium bicarbonate + 0.05 mM 2-mercaptoethanol. When macrophage differentiation was necessary, the cells were treated for 24 h with 25 nM phorbol myristate acetate (PMA) to produce an adherent culture. HEK-Blue IL-1β cells were grown in adherent culture in Dulbecco’s Modified Eagle Medium (DMEM) with L-glutamine (Gibco) + 10% FBS (heat inactivated) + 200 μg/mL HygromycinB (InvivoGen) + 100 μg/mL Zeocin (Invitrogen). All cells were cultured in a 37 °C incubator with 5% CO_2_. Adherent cells were split every 3–4 days at ~80% confluency, and 0.25% trypsin/2.21mM EDTA (Corning) were used to detach DH82 and VK2 cells for passaging. HEK-Blue IL-1β indicator cells were detached with calcium- and magnesium-free sterile PBS.

### 2.4. Human Blood Collection and Neutrophil Purification

Under approval from UC San Diego and Cedars-Sinai Medical Center Institutional Review Boards (Protocol # 131002), venous blood was obtained after informed consent from healthy adult volunteers, with heparin as an anticoagulant for whole blood and neutrophil studies. Neutrophils were isolated as described previously [36] using PolymorphPrep (Axis-Shield) to create a density gradient by centrifugation according to the manufacturer’s instructions.

### 2.5. Bacterial Killing Assays

Bacterial killing assays were modified from previous work [35,36,37]. For DH82 and THP-1 killing assays, cells were plated in 96-well plates at 3 × 10^4^ cells per well. THP-1 cells were differentiated to macrophages, as described above. The following day *S. canis* was diluted in PBS and added to the macrophages at multiplicity of infection (MOI) = 1. The culture plates were centrifuged for 5 min at 300× *g* to facilitate bacterial contact, and then plates were incubated at 30 min at 37 °C in 5% CO_2._ At the end of incubation, the supernatant was removed, and the macrophages were rinsed once with PBS before being lysed with water, serially diluted, and plated on THB agar. For human neutrophil killing assays, neutrophils were diluted to 2 × 10^6^ cells/mL in RPMI-1640 and seeded at 2 × 10^5^ cells/well in 96-well tissue culture plates. *S. canis* was diluted in RPMI-1640 was added to neutrophils at MOI = 1. Plates were centrifuged at 300× *g* for 5 min to facilitate bacterial contact with neutrophils, and then incubated at 37 °C in 5% CO_2_ for 30 min or 60 min. Samples were lysed with water, serially diluted, and then plated on THB agar. For human and murine whole-blood killing assays, 90 µL of whole blood (peripheral blood from human venipuncture or murine cardiac puncture) and 10 µL containing 1 × 10^5^ colony-forming units (CFU) of *S. canis* were incubated at 37 °C with rotation for 30 min or 60 min, and plated on THB agar. In all assays, *S. canis* survival was calculated as a percentage of the inoculum.

### 2.6. Reactive Oxygen Species Assays

Induction of reactive oxygen species release from DH82 cells was adapted from previous work [37]. Briefly, DH82 cells in confluent adherent culture were dissociated and washed in calcium and magnesium-free HBSS. The cells were stained with 2′,7′-dichlorofluorescein diacetate (Sigma Aldrich), seeded into 96-well culture plates, and infected with *S. canis* at MOI = 10 suspended in HBSS with calcium and magnesium. Plates were incubated at 37 °C with 5% CO_2_ for 120 min, and every 20 min, fluorescence intensity (485 nm excitation/530 nm emission) was measured using an EnSpire Multimode Plate Reader (PerkinElmer). Samples were normalized to fold change of fluorescence signal of time = 0.

### 2.7. IL-1β Induction Assays

Detection of THP-1 cell IL-1β release was measured as adapted from prior work [38]. HEK-Blue IL-1β reporter cells (50,000 cells per well in 96-well plates) were stimulated for 16 h at 37 °C in 5% CO_2_ with 50 μL of supernatants from THP-1 macrophages previously infected with *S. canis* or *S. pyogenes* at MOI = 1 for 30 min. After 18 h, supernatants from the HEK-Blue cells were analyzed for secreted alkaline phosphatase activity by the addition of 50 μL of HEK-Blue supernatants onto 150 μL of Quanti-Blue reagent (Invivogen) and monitoring the optical density at 620 nm via an EnSpire Multimode Plate Reader. Four independent replicate experiments were performed, and data compiled and expressed as relative units normalized to the mean optical density for the GAS group across all four experiments.

### 2.8. Adherence Assays

Vaginal epithelial adherence assays were performed as adapted from prior methods [35,39]. VK2 cells were grown to confluency in 24-well tissue culture plates. Once confluent, VK2 cells were infected with *S. canis* or *S. agalactiae* at MOI = 1 (assuming 1 × 10^6^ VK2 cells per well). Bacteria was brought into contact with the VK2 cells by centrifugation for 1 min at 300× *g*. Cells were incubated at 37 °C in 5% CO_2_ for 30 min, supernatant was removed, and cells washed 6X with sterile PBS. Cell layers were incubated for 5 min with 100 μL 0.25% trypsin/2.21mM EDTA after which 400 μL of 0.025% Triton-X in PBS was added. Wells were mixed vigorously 30X to ensure detachment and lysis, and bacterial recovery was determined by plating on THB agar. Data was expressed as a percentage of adherent CFU compared to original inoculum.

### 2.9. Human Sera Titer Assays

For detection of human titers against SCM, a purified truncated form of SCM (KO173225) [31] which does not bind human IgG Fc, was immobilized on 96-well high-binding microtiter plates (Corning Cat# 3361) at 1 µg/well via overnight incubation at RT. Wells were washed 3X with PBS + 0.05% Tween 20 and blocked with 1X Reagent Diluent (from R&D Systems, Cat#841380) for 1 h at RT. Twenty human serum samples were diluted 1:100, 1:1000, and 1:10,000 in Reagent and added at 100 uL/well. As a positive control, recombinant human IgG (BioRad Cat# HCA192) was added at 0.5 ug, 5 ng, and 50 pg/well in place of SCM protein. Negative controls included SCM-coated wells incubated with Reagent Diluent. Diluted serum and controls were incubated for 2 h at 37 °C and washed 3X with PBS + 0.05% Tween 20. Human serum binding was detected using a Goat anti-human Ig AF488 (diluted to 1:500, Southern Biotech, Cat#2010-30) and incubated for 1 h at 37 °C in the dark. Wells were washed 1X with PBS + 0.05% Tween 20 and 1X with PBS alone to remove residual Tween 20. Fluorescence (485 nm excitation/530 nm emission) was detected an EnSpire Multimode Plate Reader and data expressed at relative fluorescence units.

### 2.10. Animals

The UCSD Institutional Animal Care and Use Committee (Protocol #S00227M) approved all animal protocols and procedures. Wildtype (WT) CD-1 male and female mice aged 8-10 weeks were purchased from Charles River Laboratories (strain code 022). Groups were assigned randomly and housed at 5 animals per cage in separate cages. Mice were allowed to eat and drink *ad libitum*.

### 2.11. In Vivo Intradermal Infection Model

For intradermal infection models adapted from prior work [30], CD1 male and female mice (n = 20/group) were shaved 1 d prior to infection. On each flank, mice were injected subcutaneously with 100 µL of PBS containing either 1 × 10^8^ CFU WT *S. canis* strain G361 or Δ*scm* mutant. Sides receiving WT or Δ*scm* (left and right) were alternated at random across groups of mice. Lesions were imaged daily for three days and surface area calculated using ImageJ software.

### 2.12. In Vivo Sepsis Model

The UCSD Institutional Animal Care and Use Committee (Protocol #S00227M) approved the anticipated mortality and study design. For in vivo survival studies, CD-1 male and female mice (n = 10/group) were intraperitoneally (i.p.) injected with 100 µL of PBS containing 5 × 10^7^ CFU wildtype GAS strain 5448, WT *S. canis* strain G361 or Δ*scm* mutant. Mice were monitored three times daily for mortality. Analgesics were not administered during systemic infection due to potential effects on the study outcome.

### 2.13. In Vivo Vaginal Colonization Model

For vaginal colonization models, CD1 females (n = 18/group in single challenge experiments or n = 20 in competition experiments) were treated i.p. with 0.5 mg β-estradiol in 100 μL sesame oil (5 mg/mL) to synchronize estrus as described previously [40]. After 24 h, mice were vaginally inoculated with 1 × 10^7^ CFU WT GBS COH1, WT *S. canis* strain G361 or Δ*scm* mutant suspended in 10 µL of PBS. For WT *S. canis* strain G361 and Δ*scm* competition experiments, mice were vaginally inoculated with 1 × 10^7^ CFU each of G361 and Δ*scm* suspended in 10 µL of PBS. Colonization was monitored daily by collecting vaginal swabs (Puritan, Cat. # 25-801 A 50). Bacterial load was determined by serial dilution plating on CHROMagar™ StrepB base (DRG International Inc) and where necessary, WT G361 and Δ*scm* distinguished by plating on CHROMagar containing erythromycin (5 µg/mL).

### 2.14. Flow Cytometry

Vaginal swab samples obtained during swabbing for bacterial colonization were subjected to flow cytometry as adapted from previous work [36,41]. Vaginal lumen cells were released from vaginal swabs by vortexing for 2–3 s, passed through a 40-μm-pore-size filter, and pelleted at 500× *g* for 5 min. Cells were blocked with 1:200 mouse BD Fc-block (BD Biosciences) for 15 min on ice in 50 µL of PBS containing 1 mM EDTA, 1% FBS, and 0.1% sodium azide. Cells were stained for surface markers using the following antibodies at 5 μg/mL for 30 min on ice: anti-CD11b-fluorescein isothiocyanate (FITC) (clone M1/70, catalog no. 553310; BD Pharmingen), anti-c-kit-phycoerythrin (PE) (clone 2B8, catalog no. 1880-09; Southern Biotech), anti-CD8-PerCP-Cy5.5 (where PerCp is peridinin chlorophyll protein) (clone 53–6.7, catalog no. 100734; Biolegend), anti-F4/80-PE-Cy7 (clone BM8, catalog no. 123114; Biolegend), anti-Ly6G-allophycocyanin (APC) (clone 1A8, catalog no. 127614; BioLegend), anti-MHC-II-APC-Fire750 (clone M5/114.15.2, catalog no. 107652; BioLegend), anti-FcεRI-Pacific Blue (clone MAR-1, catalog no. 134313; BioLegend), and anti-CD45-BV510 (clone 30-F11, catalog no. 103138; BioLegend). Samples were washed 3X in PBS containing 1 mM EDTA, 1% FBS, and 0.1% sodium azide. Samples were run on a BD FACSCanto II (BD Biosciences), were gated on unstained cells, and positive signals were determined using single-stain controls. Data were analyzed with FlowJo, version 10.2, software (FlowJo LLC).

### 2.15. Tissue Histology

Whole reproductive tract tissues were collected at day 3 post-inoculation, fixed in 10% neutral buffered formalin for 24 h, and dehydrated by an ethanol gradient and embedded in paraffin. Tissue sectioning and hematoxylin and eosin (H&E) staining was performed by the UC San Diego Comparative Phenotyping Core. H&E-stained slides were examined for presence and character of inflammation by an ACVP board-certified veterinary anatomic pathologist. Representative images were captured using a Leica brightfield microscope and color CCD camera.

### 2.16. Statistical Analyses

In vitro and ex vivo experiments were repeated at least three times independently with at least three technical replicates with the exception of human serological studies which were performed in technical duplicate and analyzed twice independently. Mean values from independent experiments were used to represent biological replicates for statistical analyses. In vivo experiments were conducted at least twice independently which each mouse serving as a biological replicate. Experimental data was combined prior to statistical analyses. Data sets were subjected to D’Agostino & Pearson normality test to determine Gaussian distribution before selecting the appropriate parametric or non-parametric analyses. In instances where experimental numbers (n) were too small to determine normality (Figure 1C, Figure 2A,C–F, Figure 3A,B and Figure 4A–C,E) data were assumed non-parametric. Analyses include parametric test two-way ANOVA with Sidak’s multiple comparisons post-test, and non-parametric tests including Kruskal Wallis test with Dunn’s multiple comparisons post-test, Friedman test with Dunn’s multiple comparisons test, Wilcoxon matched-pairs signed rank test with Spearman rank-order correlation, and Log-rank (Mantel-Cox) test as indicated in figure legends. Statistical analyses were performed using GraphPad Prism, version 8.4.3 (GraphPad Software Inc., La Jolla, CA, USA). *p* values < 0.05 were considered statistically significant.

## 3. Results

### 3.1. SCM Deficiency Minimally Impacts S. canis Growth, Biofilm Formation, Hemolytic Activity, and Sensitivity to Aminizing and Oxidizing Agents

The M-protein is a well-characterized virulence factor of *S. pyogenes*, and the orthologous nature of the SCM protein suggests a potential for a similar role in *S. canis* pathogenesis. To investigate the role of SCM in *S. canis* virulence, we utilized a SCM insertional mutant made from the *S. canis* G361 clinical isolate strain that no longer produces the SCM protein and exhibits reduced aggregate formation in culture [32]. We confirmed that loss of SCM did not alter in vitro growth of *S. canis* compared to wildtype *S. canis* in bacteriologic media (THB, Figure 1A) or tissue culture media (RPMI-1640, Figure 1B). Since M protein types have been associated with biofilm formation in *S. pyogenes* [42], we assessed the contribution of SCM to biofilm formation in *S. canis.* No differences in biofilm formation between WT G361 and Δ*scm* were observed in RPMI-1640 media, however, in THB media, mutant strains had decreased biofilms relative to WT as detected by fluorescence (Figure 1C). This finding mirrored what was observed in *S. pyogenes* 5448 and Δ*emm1*. Microscopically, WT G361 formed similar biofilms in THB and RPMI-1640 media, however, Δ*scm* displayed decreased bacterial aggregation in RPMI-1640 compared to WT G361 (Figure 1D). Loss of SCM did not impact beta-hemolytic activity on blood agar (Appendix A). Sensitivity to aminizing or oxidizing agents (hypochlorite and hydrogen peroxide, respectively) was not decreased due to SCM-deficiency (hypochlorite MIC: 0.26 mM and hydrogen peroxide MIC: 0.83 mM for both strains in three independent experiments).

### 3.2. SCM Does Not Contribute to S. canis Survival, Reactive Oxygen Species Release, or Cytokine Production in Macrophages

The M-protein of *S. pyogenes* contributes to resistance of host immune cell-mediated killing [43,44]. To investigate if the SCM protein of *S. canis* shares these immunomodulatory functions, we conducted bacterial killing and immune response assays in vitro using a variety of immune cell lines as well as whole blood. Using the canine macrophage cell line DH82, we observed no differences in bacterial killing (Figure 2A) or induction of reactive oxygen species (Figure 2B) between WT and SCM-deficient *S. canis*. Using the human monocyte cell line THP-1, we observed no differences between differentiated THP-1 cell control of WT and SCM-deficient *S. canis,* or *S. pyogenes* WT strain 5448 (Figure 2C). Since *S. pyogenes* M-protein activates the NLRP3 inflammasome [38], we measured levels of activated IL-1β released by infected THP-1 cells using an IL-1β HEK-Blue reporter cell assay. Although WT 5448 infection induced twice as much active IL-1β compared to isogenic Δ*emm1* as observed previously [38], we detected no difference in IL-1β induction between the WT G361 and Δ*scm* conditions, and levels were similar to that induced by the Δ*emm1* strain (Figure 2D).

### 3.3. SCM Deficiency Does Not Alter S. canis Susceptibility to Human Whole Blood and Neutrophils nor Is SCM Antisera Detected in Healthy Human Samples

To assess *S. canis* virulence in more clinically relevant conditions, we gauged survival of *S. canis* exposed to human whole blood or purified neutrophils isolated from peripheral human blood. We did not detect any differences between WT G361 and Δ*scm* survival in whole blood, and recovered greater than 100% of the bacterial inoculum in the majority of donors suggesting *S. canis* is not readily killed in human blood (Figure 2E). Additionally, we did not observe any differences in WT G361 and Δ*scm* exposed to primary human neutrophils (Figure 2F), although bacterial survival across conditions was <10%, suggesting human neutrophils demonstrate more potent killing than either the canine or human monocyte/macrophage cells lines (Figure 2A,C). The *S. pyogenes* M proteins are widely recognized as immunodominant antigens [45], and feline antisera against SCM has been reported [46]. To investigate whether anti-SCM IgG titers are present across healthy human donors, we screened 20 human donors for IgG binding to a purified truncated form of SCM (KO173225) [31] which does not bind human IgG Fc. Human recombinant IgG immobilized on the microtiter plates was used as a positive control. Bound IgG was only detected at 1:100 serum dilutions and did not achieve the level of the positive controls at any dilution, suggesting signal is likely due to background (Figure 2G).

### 3.4. S. canis Is Highly Virulent in Mouse Models of Systemic and Dermal Infection, yet SCM Does Not Contribute to Virulence in These Models.

Since in vitro assays do not fully reflect complex host-microbe interactions, and thus might not be sensitive enough to assess the subtler contributions of SCM to *S. canis* virulence, we undertook mouse models of invasive disease. Initial studies with murine whole blood revealed no differences between WT G361 and Δ*scm* survival, and we recovered more than 200% of the bacterial inoculum in the majority of samples suggesting *S. canis* is not readily killed in mouse blood (Figure 3A). To interrogate whether SCM contributes to *S. canis* virulence in soft tissue infection, CD1 mice received intradermal injection of 1 × 10^8^ CFU of WT G361 or Δ*scm* and lesion size monitored over 3 days. Although both WT G361 and Δ*scm* led to visible formation of skin lesions, no difference in lesion size was observed at any time point (Figure 3B,C). To assess whether SCM contributes to *S. canis* morbidity in systemic infection, CD1 mice received intraperitoneal injection with 5 × 10^7^ CFU of WT G361, Δ*scm*, or WT *S. pyogenes* 5448 and mortality monitored over 3 days. Infection with WT G361 or Δ*scm* resulted in rapid decline with 100% mortality by 18 h post-infection. In contrast, *S. pyogenes* 5448 exhibited slower mortality although not statistically significant (*p* = 0.096, Figure 3D).

### 3.5. S. canis Adheres to Vaginal Epithelial Cells and Persists in a Murine Model of Vaginal Colonization, and SCM Confers a Fitness Advantage in This Environment

*S. canis* is commonly isolated from the urogenital tract of dogs [47], and the G361 strain used in these studies was originally isolated on a vaginal swab from a woman who suffered premature membrane rupture during pregnancy [33]. Since the majority of *S. canis* strains are SCM positive [3], we hypothesized that the SCM protein may provide a fitness advantage to *S. canis* in the urogenital tract. To study the potential role of SCM in vaginal colonization, we first assessed the adherence of WT G361 or Δ*scm* to immortalized human vaginal epithelial cells (VK2) and included a strain of the common vaginal colonizing bacterial species group B *Streptococcus* (GBS) as a comparison. Although no differences in vaginal cell adherence was observed between WT G361 and Δ*scm,* both *S. canis* isolates adhered significantly better than GBS (*p* = 0.016 and 0.048 respectively, Figure 4A).

To assess whether SCM contributes to *S. canis* vaginal persistence in vivo, we established a novel mouse model of vaginal colonization using the wildtype and SCM mutant strains of the G361 isolate. Adapted from previous mouse models of GBS vaginal colonization [40], female CD1 mice were synchronized for estrous stage, and inoculated with a single vaginal dose of 1 × 10^7^ CFU of WT G361, Δ*scm*, or GBS, and bacterial burdens monitored via vaginal swabbing daily for 3 days. Mice inoculated with WT G361 exhibited significantly higher bacterial burdens than Δ*scm* (*p* = 0.013) or GBS (*p* = 0.003) on day 1 post-inoculation, but no significant differences were seen in subsequent time points (Figure 4B). To determine if *S. canis* induced a local vaginal immune response, cells recovered from day 3 swabs from WT G361, Δ*scm*, or PBS-inoculated control mice were stained and analyzed via flow cytometry with antibodies for the following cell surface markers: CD45, CD11b, CD8, major histocompatibility complex class II (MHC-II), F4/80, Ly6G, FcεRI, and c-kit. CD45+ cells were analyzed for the presence of additional surface markers and total cell counts of each sub-category were reported. CD8+ T cells (CD45+ CD8+), mast cells (CD45+ c-kit+ FcεRI+), macrophages (CD45+ CD11b+ F4/80+), neutrophils (CD11b+ Ly6G+), and antigen presenting cells (CD11b+/- MHC-II+) were all observed, but numbers did not significantly differ across groups (Figure 4C). Additionally, we collected reproductive tract tissues at day 3 post-inoculation. Histological examination of the vaginal epithelium showed a range of appearances consistent with estrus (keratinized epithelium, Figure 4D, top images) or metestrus (neutrophil infiltration, Figure 4D, bottom images), no differences were observed across treatment groups. To determine whether SCM-deficiency confers a disadvantage to *S. canis* vaginal colonization, female CD1 mice were vaginally inoculated with equal amounts of WT G361 and Δ*scm* (1 × 10^7^ CFU of each) and bacterial burdens quantified over 3 days via vaginal swab. WT G361 and Δ*scm* were differentiated by plating on agar with and without erythromycin. In competition, Δ*scm* showed significantly lower burdens than WT G361 on days 2 and 3 post-inoculation. To confirm that the Δ*scm* mutant was not losing the plasmid insertion in vivo over time, we assessed the stability of erythromycin resistance in CD1 mice singly challenged with Δ*scm* by plating on both selective and non-selective agar plates. No differences were observed in bacterial counts using either erythromycin selection or no selection (Appendix A).

## 4. Discussion

*S. canis* is a well-known pathogen of dogs, cats, and other mammals, and an opportunistic zoonotic disease of humans [48,49,50,51,52], yet molecular factors promoting colonization and virulence are poorly understood. SCM is hypothesized to have similar roles as the diverse M-like proteins of other streptococcal species in the context of pathogenesis and immune evasion. To this point, multiple studies demonstrate SCM interactions with host proteins [29,31,53]; however, to date, studies have not investigated the role that SCM plays in pathogenesis and colonization in vivo. Our work seeks to bridge this gap in knowledge and define the importance of SCM in epithelial and immune cell interactions in vitro and in novel mouse models of *S. canis* pathogenesis and colonization. Overall, we observed minimal contributions of SCM to *S. canis* interactions with the host in either commensal or pathogen contexts using the *S. canis* strain G361 which possesses a SCM type 2, group 1 allele [12]. Importantly, it remains possible that SCM proteins belonging to different group alleles may differentially impact *S. canis* interactions with its host. Prevalence of SCM in veterinary *S. canis* isolates is quite high: both colonizing and invasive isolates are 70.6-90.0% SCM positive by PCR [3,54] suggesting possible biological pressure(s) to retain SCM for successful *S. canis* colonization and/or invasive disease.

In terms of bacterial characteristics, SCM is non-essential for growth, either in nutrient-rich bacteriologic media or nutrient-poor tissue culture media. SCM also does not contribute to hemolysis in *S. canis*, similar to reports for other streptococcal M-like proteins [55]. In *S. pyogenes*, M1 over-expression enhances biofilm formation and M1-deficient strains demonstrated reduced biofilm formation in bacteriologic media [42]. Similarly, we noted decreased biofilm formation in SCM-deficient *S. canis* in bacteriologic media, however, this finding was not observed in conditions reflective of the host environment (RPMI, Figure 1C). In *S. pyogenes*, M protein contributes to the hydrophobicity of the bacterial surface, leading to greater biofilm formation [42]. Thus, loss of SCM in *S. canis* possibly reduces surface proteins with greater hydrophobicity, leading to the observed decrease in biofilm formation. Alternatively, because SCM activity is associated with *S. canis* aggregation microcolony formation in vitro [56], loss of homophilic interactions between SCM in the Δ*scm* strain may alter the morphology of *S. canis* biofilms [32,57]. In fact, although there was no difference in the relative fluorescent units (RFU) of WT G361 and Δ*scm* biofilms in RPMI-1640, the morphology of the biofilms was visibly different. While WT G361 biofilms grown in RPMI-1640 formed microcolonies comparable to WT G361 grown in THB, Δ*scm* grown in RPMI-1640 formed diffuse biofilms with fewer microcolonies (Figure 1D). The impact of this altered biofilm morphology on *S. canis* interactions within different host tissues is a topic of future study.

Streptococcal M proteins play key roles in virulence and host immune evasion. *S. pyogenes* M proteins confer resistance against phagocytosis by inhibiting the alternate complement pathway [58,59], or alternatively, through forming bacterial aggregates with self [56,60] or as a complex with host cells [61,62]. Similarly, in *S. canis*, this self-aggregation is lost in the absence of SCM [32]. However, in our assays with canine or human macrophage cell lines, we found that presence of SCM did not alter macrophage-mediated killing of *S. canis* (Figure 2). In fact, neither WT or Δ*scm* bacteria were effectively controlled by macrophage cell lines or human and mouse whole blood, with ~100% or more of inoculum recovered at the experimental end point. This contrasts our findings with isolated human neutrophils, which efficiently reduced *S. canis* viability by more than 90% over the assay (Figure 2F), yet with no significant contribution of SCM to *S. canis* neutrophil resistance. This observation is in line with a previous study which did not observe any differences in neutrophil control of SCM-positive or SCM-negative *S. canis* except for when exogenous SCM and/or plasminogen was present [57].

Another notable activity of M-like proteins is resistance to host immune defense molecules and activation of innate inflammatory responses, best characterized in *S. pyogenes*. *S. pyogenes* M1 neutralizes cathelicidin [30,34], stimulates neutrophil recruitment and myeloperoxidase release [63], and activates the NLRP3 inflammasome triggering IL-1β-dependent pyroptosis in macrophages [38]. In our in vitro immune cell models, we observed no differences in reactive oxygen species (ROS) production or induction of IL-1β between SCM-deficient and wildtype *S. canis* (Figure 2). Overall, we observed only modest induction of ROS, and minimal induction of IL-1β in macrophages infected with *S. canis* with levels similar to that of M protein-deficient *S. pyogenes*. This failure to elicit an immune response could benefit *S. canis* as it avoids the production of inflammatory cytokines and pyroptosis that have been shown to be critical for the control of some bacterial pathogens [64]. However, in our in vivo models of *S. canis* invasive infection, we did not distinguish distinctive contributions of SCM. No differences in skin lesion sizes were observed between SCM-deficient and WT *S. canis,* and in general, we observed smaller lesion sizes than that reported for a similar model in *S. pyogenes* [30]. Time points beyond 3 days were not assessed in our studies. We observed high lethality of *S. canis* compared to *S. pyogenes* in a murine model of systemic infection at a dose of 5 × 10^7^ CFU, similar to a previous study, which demonstrated an LD90 of 3 × 10^7^ CFU in mice [28].

Additionally, the antigenic nature of *S. pyogenes* M proteins is well-recognized [65,66], and anti-M protein responses are associated with host protection against repeat infection by strains of the same M type [67,68]. Regarding *S. canis*, antisera against SCM has been detected in diseased cats [46]. No compelling evidence for SCM anti-sera in healthy human donors was detected in our cohort, however, subjects with known active or prior *S. canis* infections or exposure were not included in this study.

Along with relevance to pathogenesis, streptococcal M proteins play prominent roles in host colonization. This phenomenon may be in part due to mediating adherence [69], although M proteins may not necessarily promote adhesion to all epithelial types [56], and expression of certain M proteins may even reduce epithelial adherence [70]. Although we observed no contribution of SCM in adherence to human vaginal epithelial cells in vitro, it is possible that these cells do not produce the same extracellular matrix or surface proteins to recapitulate in vivo conditions. Furthermore, we observed a striking increase is S. canis adherence to VK2 compared to the frequently isolated vaginal bacterium group B Streptococcus (Figure 3) [71,72] but the underlying molecular basis for this heightened adherence is currently unknown. In vivo, *S. canis* persisted in the mouse vaginal tract at similar levels to GBS highlighting the utility of this murine model in studying *S. canis* factors contributing to urogenital colonization. We detected no changes to immune cell profiles or epithelial appearance suggesting that *S. canis* does not stimulate a robust vaginal immune response soon after introduction. This contrasts with other human pathogen vaginal colonization models which do invoke immune responses [72,73,74] and may reflect the primarily colonizing role for *S. canis* in mammals such as cats and dogs. Although we observed minimal contribution of SCM in *S. canis* colonization in a single bacterial challenge model, when both WT *S. canis* and Δ*scm* were introduced in competition, WT *S. canis* demonstrated a competitive advantage (Figure 4E). The underlying mechanisms for this SCM-mediated advantage in niche establishment are unknown, but this finding provides rationale for the conservation and heterogenetic variability of SCM across clinical isolates [12,23].

In summary, we have deployed several new in vitro and mouse models of S. *canis* infection and colonization and interrogated the role of SCM in these models using a targeted knockout mutant. We observed minimal impact of SCM deficiency in invasive disease models, but found an SCM-associated advantage in vaginal colonization. These results suggest that the role of SCM is distinct from the repertoire of virulence mechanisms ascribed to other streptococcal M proteins. Further studies are necessary to determine the mechanisms underlying decreased colonization fitness of SCM-deficient *S. canis* and to extend our findings to other SCM variants. Identification of these mechanisms will provide insight into the viability of SCM as a vaccine target for the zoonotic pathogen *S. canis*.

## Figures and Tables

**Figure 1 microorganisms-09-00183-f001:**
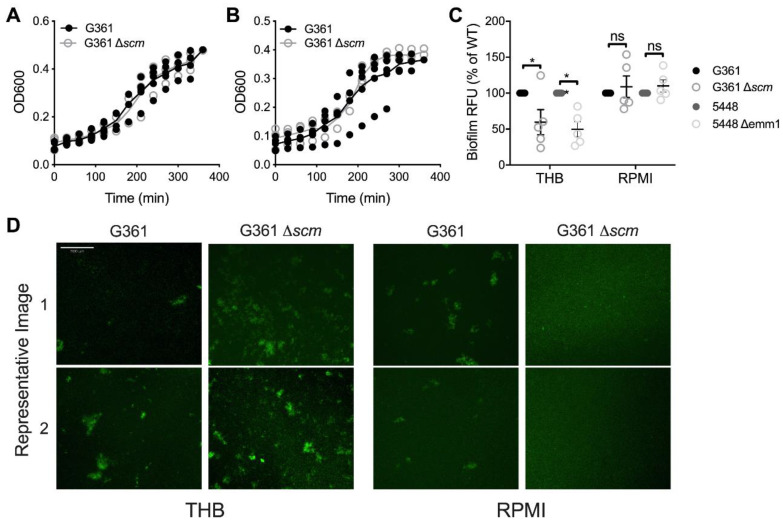
SCM deficiency minimally impacts *S. canis* growth, hemolytic activity, and biofilm formation. Growth curves of *S. canis* G361 or G361Δ*scm* in Todd-Hewitt broth (THB, (**A**)) or RPMI-1640 (RPMI, (**B**)) as measured by optical density (OD_600_). (**C**) Biofilm formation of *S. canis* G361 or G361Δ*scm* or GAS 5448 or 5448Δ*emm1* in THB or RPMI quantified by SYTO 13 fluorescence and expressed as the percent fluorescence of the WT strain. (**D**) Representative images (two per condition) of *S. canis* G361 or G361Δ*scm* biofilms grown for 48 h in THB or RPMI and stained with SYTO 13. Symbols represent individual experimental replicates (**A**–**C**) with lines indicating interquartile ranges. Representative images are of independent experimental replicates, scale bar = 200 μm (**D**). Data were analyzed by two-way ANOVA with Sidak’s multiple comparisons post-test (**A**–**C**). *, *p* < 0.05.

**Figure 2 microorganisms-09-00183-f002:**
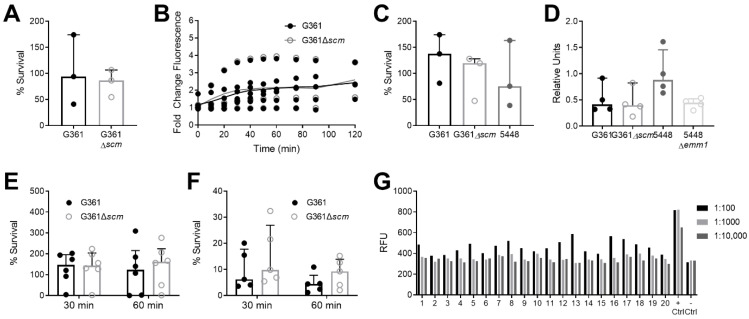
SCM does not alter *S. canis* survival, reactive oxygen species release, cytokine production, nor induce antigenic activity in human sera. (**A**) Percent survival of *S. canis* G361 or G361Δ*scm* after 30 min of exposure to canine DH82 macrophages, MOI = 1. (**B**) Reactive oxygen species production by DH82 macrophages infected with *S. canis* G361 or G361Δ*scm,* MOI = 10, and normalized to uninfected cells. (**C**) Percent survival of *S. canis* G361 or G361Δ*scm* after 30 min of exposure to human THP-1 differentiated macrophages, MOI = 1. (**D**) THP-1 cells were infected with *S. canis* G361 or G361Δ*scm,* MOI = 1, and cell supernatant added to HEK-Blue cells. Alkaline phosphatase activity was measured colorimetrically at OD620 and background signal (uninfected cell supernatant) was deducted. Fold IL-1β release was calculated versus GAS across four independent experiments. (**E**) Percent survival of *S. canis* G361 or G361Δ*scm* after 30 or 60 min of infection in human whole blood. (**F**) Percent survival of *S. canis* G361 or G361Δ*scm* after 30 or 60 min of infection in isolated human neutrophils, MOI = 1. (**G**) Quantification of human IgG titers, expressed as relative fluorescent units (RFU), for a purified truncated form of SCM (*n* = 20 donors) via modified ELISA using diluted human sera, positive control: recombinant human IgG, negative control: buffer only. Symbols represent independent experimental replicates (**A**–**D**), biological replicates ((**E**), *n* = 6/group, (**F**), *n* = 5/group), or the results of one independent experiment (**G**), performed twice independently), with lines indicating medians and interquartile ranges. Data were analyzed by Wilcoxon matched-pairs signed rank test (**A**), two-way ANOVA with Sidak’s multiple comparisons post-test (**B**,**E**,**F**), or Friedman test with Dunn’s multiple comparisons test (**C**,**D**) and determined not significant.

**Figure 3 microorganisms-09-00183-f003:**
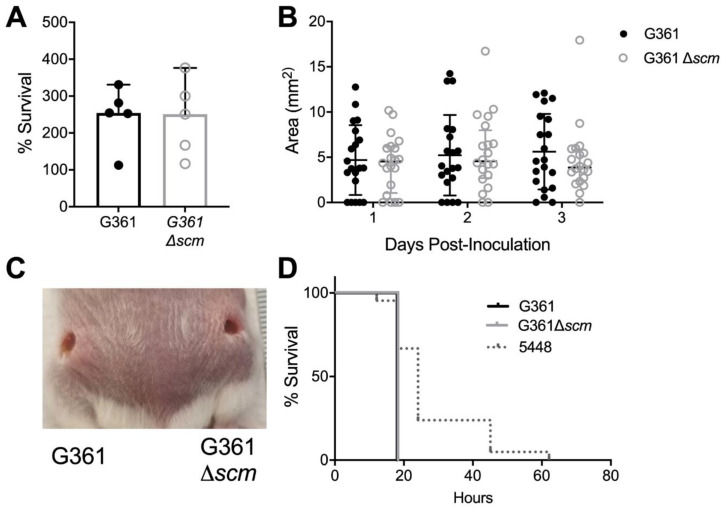
*S. canis* is highly virulent in mouse models of systemic and dermal infection, yet SCM does not contribute to virulence in these models. (**A**) Percent survival of *S. canis* G361 or G361Δ*scm* after 30 min of infection in murine whole blood collected from CD1 mice. (**B**) CD1 male and female mice were infected subcutaneously with 1 × 10^8^ CFU of WT *S. canis* G361 or G361Δ*scm* and skin lesion size measured daily. (**C**) Representative image of skin lesions three days post subcutaneous infection with WT *S. canis* G361 (left) or G361Δ*scm* (right). (**D**) CD1 male and female mice were infected intraperitoneally with 5  ×  10^7^ CFU of WT *S. canis* G361, G361Δ*scm*, or *S. pyogenes* 5448 and survival monitored over 3 days. Symbols represent biological replicates ((**A**), *n* = 5/group, (**B**), *n* = 20/group, and (**D**), *n* = 10-21/group) with lines indicating medians and interquartile ranges (**A**,**B**) or percentage survival (**D**). Data were analyzed by Wilcoxon matched-pairs signed rank test (**A**), two-way ANOVA with Sidak’s multiple comparisons post-test (**B**) or Log rank Mantel-Cox test (**D**) and determined not significant.

**Figure 4 microorganisms-09-00183-f004:**
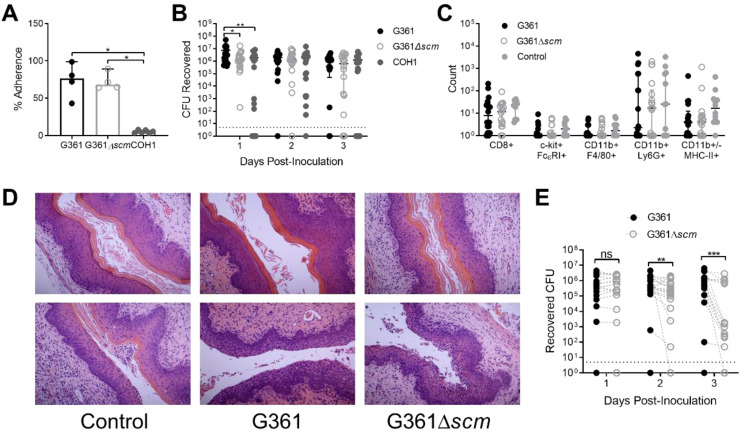
*S. canis* adheres to vaginal epithelial cells and persists in a murine model of vaginal colonization, and SCM confers a fitness advantage in this environment. (**A**) Percent adherence of *S. canis* G361, G361Δ*scm*, or GBS COH1 to VK2 cells after 30 min of infection, MOI = 1. CD1 female mice were vaginally administered 1  ×  10^7^ CFU of WT *S. canis* G361, G361Δ*scm*, or WT GBS COH1, or PBS as a control. (**B**) Mice were vaginally swabbed daily, and the levels of bacterial CFU recovered from swabs are shown. (**C**) Cells collected from day 3 post-inoculation were analyzed for surface markers via flow cytometry. Total cell counts of each population recovered on the swabs are shown. (**D**) Vaginal epithelial tissues were fixed, sectioned, and stained with H&E. Histological examination revealed keratinized epithelium (top images) and neutrophil infiltration (bottom images) similarly across treatment groups. Magnification = 200X. (**E**) CD1 female mice were vaginally administered 1  ×  10^7^ CFU each of WT *S. canis* G361 and G361Δ*scm* in competition. Mice were vaginally swabbed daily, and the levels of bacterial CFU recovered from swabs are shown. Symbols represent biological replicates ((**B**), *n* = 18/group, (**C**), *n* = 12–18/group, and (**E**), *n* = 20/group) or the means of four independent experimental replicates (**B**), performed in technical duplicate) with lines indicating medians and interquartile ranges. Dotted line in (**B**,**E**) indicates limit of detection. Data were analyzed by Kruskal-Wallis test with Dunn’s multiple comparisons test (**A**), two-way ANOVA with Sidak’s multiple comparisons post-test (**B**,**C**) or Wilcoxon matched-pairs signed rank test (**E**). ***, *p* < 0.001; **, *p* < 0.01; *, *p* < 0.05; ns, not significant.

## Data Availability

The data presented in this study are available in Appendix A Raw Data Files here.

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
