# Peer review of "Novel Models of Streptococcus canis Colonization and Disease Reveal Modest Contributions of M-Like (SCM) Protein"

_microorganisms, 2021, doi:10.3390/microorganisms9010183_

Round 1

Reviewer 1 Report

Dear authors,

I appreciate the revised version of the manuscript and the answers provided. Congratulations on your work!

Reviewer 2 Report

Dear Authors

The overall manuscript presentation was impressive and interesting. Title is clear and informative; it displays the main objective of the study. The abstract is sectioned. It contains focused background with clear objective. The literature review follows the specific aim of the study. The research methods used ensure the reliability of the obtained results. The discussion interpretations and conclusions justified by the results of the study.

This manuscript is a resubmission of an earlier submission. The following is a list of the peer review reports and author responses from that submission.

Round 1

Reviewer 1 Report

Ingird. et al. examined the role of SCM protein from S. canis in the novel experimental models  and described that the role of SCM of S.canis is not critical for vaginal colonization, growth, hemolytic activity, survival to macrophages.

Authors concluded that the role of SCM from S.canis based on the results showing no significant differences between WT and delscm. However, to reach such conclusion one must have proper positive control experimental condition. One can simply S.pyogenes WT and its mutant as author did only in Fig 2D. (There is no delemm mutant in Fig 2C. Any reason for this? ).

I suggest authors to re-submit the manuscript with the addition of all the proper control experiment in the most of results. The present version is not suitable to be published in Microorganisms.

Minor Comments:

  1. It will be better to add description of what GAS is in line 277

  1. line 366-367: Better to put in method.

Reviewer 2 Report

Dear Authors

The overall manuscript presentation was impressive and interesting. 

Title is clear and informative; it displays the main objective of the study.

The abstract is sectioned. It contains focused background with clear objective.

The literature review follows the specific aim of the study.

The research methods used ensure the reliability of the obtained results.

The discussion interpretations and conclusions justified by the results of the study.

Reviewer 3 Report

The study conducted by Cornax et al. seeks to evaluate through several models (in vitro and in vivo models) the impact of the protein SCM on S. canis G361 virulence. In general, the work is well written, and the amount of analysis performed in an attempt to elucidate such mechanisms and interactions with the host is appreciable. Amongst the results, the authors identified an “increased” biofilm formation capability of the strain G361Δscm, as well as a greater WT G361 load, was observed in vaginal swabs when compared to its SCM-deficient isogenic mutant. The main weakness of this study lies in the fact that the biofilm formation assay was evaluated only after 24 hours of incubation and was evaluated by the traditional violet-crystal method, which is mainly adopted for initial screenings. Additional answers could have been provided, for example, with a scanning electron microscopy analysis, or better supported with the quantification of sessile biofilm-embedded cells. The main concerns are reported in the attached file.
